# Probabilistic Rank and Reward: A Scalable Model for Slate Recommendation

## Abstract

We introduce **P**robabilistic **R**ank and **R**eward (`PRR`), a scalable probabilistic model for personalized slate recommendation. Our approach allows off-policy estimation of the reward in the ubiquitous scenario where the user interacts with at most one item from a slate of $K$ items. We show that the probability of a slate being successful can be learned efficiently by combining *the reward*, whether the user successfully interacted with the slate, and *the rank*, the item that was selected within the slate. `PRR` outperforms existing off-policy reward optimizing methods and is far more scalable to large action spaces. Moreover, `PRR` allows fast delivery of recommendations powered by maximum inner product search (MIPS), making it suitable in low latency domains such as computational advertising.

## 1 INTRODUCTION

Recommender systems (advertising, search, music streaming, etc.) are becoming prevalent in society helping users navigate enormous catalogs of items to identify those relevant to their interests. In practice, recommender systems must optimize the content of an entire section of the web page that the user is navigating. This section is viewed as an ordered collection (or slate) of $K$ items (Swaminathan et al., 2017; Chen et al., 2019; Aouali et al., 2021). It often takes the form of a menu and the user can choose to interact with one of its items. Both in academia and industry, A/B tests (Kohavi & Longbotham, 2017) are seen as the golden standard to measure the performance of recommender systems. A/B tests enable us to directly measure utility metrics that rely on interventions, being the slates shown to the user. However, they are costly. Thus a clear need remains for reliable offline procedures to propose candidate recommender systems and consequently reduce the cost of A/B tests.

In this work, we propose a probabilistic model called **P**robabilistic **R**ank and **R**eward (`PRR`) for large-scale slate recommendation. `PRR` addresses the following practical limitations of existing methods. **1)** Collaborative filtering and content-based recommender systems (Su & Khoshgoftaar, 2009; Lops et al., 2011) optimize *proxy* metrics of the reward. This may lead to a striking gap between their offline evaluation and the A/B test result (Garcin et al., 2014). **2)** Counterfactual estimators (Bottou et al., 2013), which are often based on inverse propensity scoring (IPS) (Horvitz & Thompson, 1952), suffer high bias (Sachdeva et al., 2020) and variance (Swaminathan et al., 2017) in large-scale scenarios. Moreover, policy learning objectives for these estimators (Swaminathan & Joachims, 2015) are mostly not suitable for slates and large action spaces. **3)** The decision rules produced by most existing methods do not fit the engineering constraints for deployment in large-scale, low-latency systems such as computational advertising; they are either expensive or intractable. To address these challenges, our paper makes the following contributions.

**1. Problem formulation:** We formalize the ubiquitous slate recommendation setting where the user is shown a slate of $K$ items and they can choose to interact with *at most* one of its items. After that, the information received consists of two signals: did the user successfully interact with one of the items? Then if an item was interacted with, which one was it? These are referred to by *reward* and *rank*, respectively. Note that it is very common in practice that the user interacts with *at most* one item in the slate. For example, in ad placement, a click on an item causes the whole slate to disappear. As a result, the user cannot click on the other items in that slate. Similarly, in video recommendation, a click on a video to watch it reloads

the homepage and changes it. Most offline reward optimizing methods do not consider this and assume that the user can interact with multiple items in the slate.

**2. Modeling the reward and rank:** We propose a probabilistic model (`PRR`) that combines both signals, the reward and rank. This is important as both contain useful information about the user interests, and discarding one of them may lead to inferior performance. Existing methods either use one signal only or partially combine the two by assuming that the reward is a function of the rank and only use the latter.

**3. Incorporating extra features:** `PRR` distinguishes between slate and item level features that contribute to an interaction with the slate and one of its items, respectively. `PRR` also incorporates that interactions can be predicted by *engagement features* that neither represent the user interests nor the recommended items. This includes the slate size and the overall level of user activity and engagement. While these features are not used in decision-making, incorporating them helps learn the user interests more accurately. Precisely, it allows differentiating between interactions that are caused by the quality of recommendations and those that happen due to overall user engagement. Existing model-based methods do not include such features.

**4. Fast decision making:** `PRR`'s decision rule is reduced to solving a maximum inner product search (MIPS). This allows fast delivery of recommendations in large-scale tasks using approximate MIPS algorithms (Shrivastava & Li, 2014; Malkov & Yashunin, 2018) at the cost of restricting the decision rule to a particular form. Most existing offline reward optimizing methods do not take into account the real-time response requirement of production systems and as such can propose expensive decision rules.

**5. Experiments:** We show empirically that `PRR` outperforms commonly used baselines and that it is far more scalable in terms of both statistical and computational efficiency.

The remainder of the paper is organized as follows. In Section 2 we review the state of the art in off-policy, or offline, reward optimizing recommendation. In Section 3, we describe the setting for slate recommendation and present our algorithm. In Section 4, we describe the baselines and present our qualitative and quantitative results showing that `PRR` has favorable performance both in terms of recommendation quality and scalability. In Section 5, we make concluding remarks and outline potential directions for future work.

## 2 Related Work

A significant part of the classic recommender systems literature is inspired by the Netflix prize (Bennett et al., 2007) which formulated recommendation as predicting item ratings in a matrix (Portugal et al., 2018). A practitioner will often work with datasets such as MovieLens (Harper & Konstan, 2015) that include neither recommendations nor rewards, but are suitable for collaborative filtering (Su & Khoshgoftaar, 2009) or content-based recommendation (Lops et al., 2011). While interesting, such datasets and problem formulations do not reflect the actual interactions between the users and the recommender systems; they are only imperfect proxies. In particular, the performance of an algorithm on these problems and datasets may be very different from its actual A/B test performance (Garcin et al., 2014, Section 5.1). Instead, *off-policy, or offline, reward optimizing recommendation* approaches aim at directly optimizing the reward using *logged data* summarizing the previous interactions of users with the existing recommender system. These are also different from on-policy, or online, approaches that we do not cover in this work. Here the reward is learned offline using logged data. In this section, we review these approaches and compare them to ours.

### 2.1 Inverse Propensity Scoring (IPS)

Here we assume that the recommender system is represented by a stochastic policy $\pi$. That is, given a user $u$, $\pi(\cdot \mid u)$ is a probability distribution over the set of items. Under this assumption, Dudík et al. (2012) used inverse propensity scoring (IPS) (Horvitz & Thompson, 1952) to estimate the reward for recommendation tasks with small action spaces. Unfortunately, IPS can suffer high bias and variance in realistic settings such as slate recommendation. This is mainly driven by two practical factors; the action space is combinatorially large and the policy used to collect data primarily *exploit* certain recommendations with minimal or no *exploration* making its support deficient (Sachdeva et al., 2020). The high variance of IPS is acknowledged and several fixes have been proposed such as clipping, and self-normalizing importance weights (Gilotte

et al., 2018). Another solution is doubly robust (DR) (Dudík et al., 2014) which combines a reward model with IPS to reduce the variance. DR relies on a reward model and `PRR` can be integrated into it.

In slate recommendation, recent works made simplifying structural assumptions to reduce the variance. For instance, Li et al. (2018) restricted the search space by assuming that items contribute to the reward individually. Similarly, Swaminathan et al. (2017) assumed that reward is additive w.r.t. unobserved and independent ranks. The independence assumption is restrictive and can be violated in many production settings. A relaxed assumption was proposed in McInerney et al. (2020) where the interaction between the user and the $\ell$-th item in the slate, $s_\ell$, depends only on $s_\ell$, $s_{\ell-1}$ and its rank $r_{\ell-1}$. This sequential dependence scheme is not sufficient to encode the ubiquitous setting where the user views the whole slate at once and interacts with *at most* one of its items. `PRR` is model-based, does not use inverse propensity scoring and is specifically designed for this ubiquitous setting.

## 2.2 Direct Methods (DM)

Direct methods learn a reward model and then use it to estimate the performance of the recommender system. Existing methods (Sakhi et al., 2020; Jeunen & Goethals, 2021) focus on single-item recommendation (slates of size 1) and neglect that the quality of recommendations might not have a strong effect on the reward. Instead, *engagement features* that are not relevant to recommendation such as the overall level of user activity and engagement may have a higher effect on the reward. `PRR` explicitly includes nuisance parameters to capture these effects independently of the quality of recommendations.

Another popular family related to direct methods is called *click (or ranking) models* (Chuklin et al., 2015). The simplest is *click-through-rate models* which defines a single parameter for the probability that an item is clicked, possibly depending on its position or the user (Joachims et al., 2017; Craswell et al., 2008). Another type is called *cascade models* (Dupret & Piwowarski, 2008; Guo et al., 2009; Chapelle & Zhang, 2009), which is suitable when items are presented in sequential order. Later, these models were extended to accommodate multiple user sessions (Chuklin et al., 2013; Zhang et al., 2011), granularity levels (Hu et al., 2011; Ashkan & Clarke, 2012), and additional user signals (Huang et al., 2012; Liu et al., 2014). Click models are often represented as graphical models and as such define dependencies manually and are not always scalable to large action spaces. Moreover, they do not incorporate extra features that are available in recommendation since they were primarily designed for search engine retrieval. Typically they are evaluated using proxy Information Retrieval metrics such as Recall@K instead of the actual reward.

## 2.3 Policy Learning

Finding an optimal policy to implement in a recommender system requires both estimating the expected reward of policies (using either direct or IPS methods above) and then searching the space of policies to find the one with the highest expected reward (policy learning). The literature mainly focuses on combining IPS with a softmax policy for single item recommendation (Swaminathan & Joachims, 2015). Extending this to slates is challenging. In fact, it is tempting to use factored softmax policies but this may cause the learned policy to recommend slates with repeated items. This was acknowledged in Chen et al. (2019) that introduced a top-K heuristic to prevent the collapse of the policy on a single item. In our case, policy learning is not needed as the optimal policy of `PRR` falls out neatly to a MIPS task due to its parametrization.

## 2.4 Decision Making

An important challenge in practice is the design of tractable decision rules that satisfy engineering constraints. Precisely, real-world recommender systems must quickly recover a slate of items $\boldsymbol{s}$ given a context vector $\boldsymbol{x}$. The optimal decision rule under either the IPS or DM formulation amounts to solving $\mathrm{argmax}_{\boldsymbol{s} \in \mathcal{S}} L(\boldsymbol{x}, \boldsymbol{s})$. Here $L(\boldsymbol{x}, \boldsymbol{s})$ might either be the reward estimate in DM or the policy in IPS. However, the space of eligible slates $\mathcal{S}$ is combinatorially large. Thus exhaustively searching the space of slates is untenable and we must resort to finding good but implementable decision rules rather than optimal ones. There are three main strategies for doing this. **(A)** Reducing the combinatorial search over $\mathcal{S}$ to a sorting operation over a set of $P$ items. **(B)** Two-stage systems that massively reduce candidate sets with a pre-filtering operation.

**(C)** Fast approximate maximum inner product search (MIPS) on which we focus in this work since it allows end-to-end optimization as opposed to the two-stage scheme. Next, we provide more detail for each solution.

**(A) Reducing the combinatorial search to sorting:** To accelerate decision making, existing methods moved the search space from the combinatorially large set of slates $\mathcal{S}$ to the catalog of items $\{1, \ldots, P\}$. This is achieved by first associating a score for items instead of slates and then recommending the slate composed of the top-K items with the highest scores. This leads to a $\mathcal{O}(P)$ delivery time due to finding the top-K items. Unfortunately, reducing a combinatorial search to a sort is still unsuitable for low-latency recommender systems with large catalogs. We present next the common solution to improve this.

**(B) Two-stage recommendation ([Borisyuk et al., 2016](#)):** Here we first generate a *small* subset of potential item candidates $\mathcal{P}_{\text{sub}} \subset \{1, \ldots, P\}$, and then select the top-K items in $\mathcal{P}_{\text{sub}}$ leading to a $\mathcal{O}(|\mathcal{P}_{\text{sub}}|)$ delivery time. This has two main shortcomings. First, the scoring model, which selects the highest scoring items from $\mathcal{P}_{\text{sub}}$, does not directly optimize the reward for the whole slate, and rather optimizes a proxy offline metric for each item individually. This induces numerous biases related to the layout of the slate such as position biases where users tend to interact more often with specific positions ([Yue et al., 2010](#)). Second, the candidate generation and the scoring models are not necessarily trained jointly, which may lead to having candidates in $\mathcal{P}_{\text{sub}}$ that are not the highest scoring items.

**(C) Maximum inner product search (MIPS):** A practical approach to avoid the candidate generation step relies on approximate MIPS algorithms. [Koch et al. (2021)](#) demonstrated the power of such algorithms in a recommendation setting. Roughly speaking, they are capable of quickly sorting $P$ items in $\mathcal{O}(\log P)$ as long as the scores of items $a \in \{1, \ldots, P\}$ have the form $\boldsymbol{u}^\top \boldsymbol{\beta}_a$. Here $\boldsymbol{u} \in \mathbb{R}^d$ is a $d$-dimensional user embedding and $\boldsymbol{\beta}_a \in \mathbb{R}^d$ is the $d$-dimensional embedding of item $a$. This allows fast delivery of recommendation in roughly $\mathcal{O}(\log P)$ instead of $\mathcal{O}(P)$ without any additional candidate generation step. `PRR` uses approximate MIPS algorithms ([Shrivastava & Li](#), 2014; [Malkov & Yashunin](#), 2018) making it suitable for extremely low-latency systems. We note that both IPS and DM can lead to a MIPS-compatible recommender system if the model (or the policy) is appropriately parametrized. However, much of the existing literature neglect this important consideration. In our case, decision-making is reduced to a MIPS task and we use fast approximate algorithms to solve it. This allows us to avoid the two-stage scheme and its limitations.

## 2.5 Summary of Limitations

Here we summarize the limitations of existing methods. **(a) Poor estimation of the reward:** this is due to the high variance and bias of IPS, the incorrect assumptions of existing DM and their potential modeling bias (e.g., only a single item is recommended, ignoring engagement features, etc.). **(b) Policy learning for single items:** extending existing approaches to slates is complicated. **(c) Slow decision making:** the real-time response requirement is not respected by most existing methods. The two-stage system is a remarkably pragmatic compromise. But it poses some challenges as we explained before. MIPS is a reliable practical alternative to two-stage systems but existing methods are usually not MIPS compatible.

# 3 ALGORITHM

`PRR` is a binary reward model that differentiates between item-level and slate-level features. The former reflects the quality of the slate as a whole while the latter is associated with the quality of individual items in the slate. This allows `PRR` to predict whether the user will interact with the slate (the reward) and which item will be interacted with (the rank). To see this, we give an example of the output of `PRR` in [Fig. 1](#). Here we imagine that a user is interested in *technology*. We show three slates of size 2. In the left panel, the slate consists of two good[1] items: *phone* and *microphone*. The model predictions $(0.91, 0.06, 0.03)$ are the probabilities for no click, click on *phone* and click on *microphone*, respectively. The probability of a click on slate *phone, microphone* is higher than the other slates and is equal to 0.09. For comparison, the panel in the middle contains a good item (*phone*) in the prime first position but the *shoe* in the second position, which is a poor match with the user interest in technology. As a consequence, the probabilities become

---

[1]Here a good item refers to a technology item.

$$\bar{R}, r_1, r_2 = (0.91, 0.06, 0.03) \qquad \bar{R}, r_1, r_2 = (0.94, 0.04, 0.01) \qquad \bar{R}, r_1, r_2 = (0.97, 0.02, 0.01)$$

**Figure 1:** Example of 3 slates of size 2 on a technology website. From left to right are good, mixed and bad recommendations. $\bar{R}, r_1, r_2$ denote the probabilities of no-click, click on the 1st and 2nd item, respectively.

$(0.94, 0.04, 0.01)$ for no click, click on *phone* and click on *shoe*. Finally, in the right panel, we show two poor items *shoe* and *pillow* resulting in the highest no-click probability 0.97.

The goal is to establish the level of association of each item (in this case *phone*, *microphone*, *shoe* and *pillow*) with a particular user interest (in this case *technology*). At first glance, analyzing logs of successful and unsuccessful recommendations is the best possible way to learn this association. However, in practice, there are numerous factors that influence the probability of a click other than the quality of recommendations. In this example, the non-click probability of the good recommendations (*phone*, *microphone*) is 0.91 (click probability of 0.09), while the non-click probability of the bad recommendations (*shoe*, *pillow*) is 0.97 (click probability of 0.03). The change in the click probability from good to bad recommendations is relatively modest at only 0.06. Thus the model must capture additional factors that influence clicks.

To account for this, `PRR` incorporates a real-world observation made by practitioners: typically the most informative features to predict successful interactions are *engagement features*. These summarize how likely the user is to interact with the slate independently of the quality and relevance of its items to the user. This includes for example the slate size, its visibility and the level of user activity and engagement. While these features are strong predictors of interactions, they do not provide any information about which items are responsible for which interactions. In contrast, the *recommendation features*, which include the user interests and the items shown in the slate, provide a relatively modest signal for predicting interactions. But they are very important for the recommendation task. Based on these observations, `PRR` leverages the engagement features to accurately learn the parameters associated with the *useful* recommendation features.

`PRR` also incorporates the information that different positions in the slate may have different properties. Some positions may boost a recommendation by making it more visible, and other positions may lessen the impact of the recommendation. To see this, consider the example in Fig. 1, the probability of clicking on *shoes* increased by 0.01 when placed in the prime first position (slate in the middle) compared to placing it in the second position (slate in the right panel).

It may be relevant to draw an analogy. Optical astronomers who take images of far-away galaxies need to develop a sophisticated understanding of many local phenomena: the atmosphere, the ambient light, etc. The understanding of all these large effects allows them to construct precise images of faint objects. Similarly, `PRR` is able to capture a weak recommendation signal by carefully incorporating the other factors that often have high contributions to predicting the reward such as the engagement features and position biases. After providing the intuition of our approach, we are in a position to formally present it. But before that, we need to introduce our setting and notation.

## 3.1 Setting

For any positive integer $P$, we define $[P] = \{1, 2, \ldots, P\}$. Vectors and matrices are denoted by bold letters. The $i$-th coordinate of a vector $\boldsymbol{x}$ is $x_i$; unless the vector is already indexed such as $\boldsymbol{x}_j$, in which case we write $x_{j,i}$. Let $\boldsymbol{A} \in \mathbb{R}^{P \times d}$ be a $P \times d$ matrix. Then for any $i \in [P]$, the $d$-dimensional vector corresponding to the $i$-th row of $\boldsymbol{A}$ is denoted by $\boldsymbol{A}_i \in \mathbb{R}^d$. Items are referenced by integers so that $[P]$ denotes the catalog of $P$ items. We define a *slate* of size $K$, $\boldsymbol{s} = (s_\ell)_{\ell \in [K]} = (s_1, \ldots, s_K)$, as a $K$-permutation of $[P]$, which is an ordered collection of $K$ items from $[P]$. The space of all slates of size $K$ is denoted by $\mathcal{S}$.

We consider a *contextual bandit* setting where the agent interacts with users as follows. The agent observes a $d_x$-dimensional *context* vector $\boldsymbol{x} \in \mathcal{X} \subseteq \mathbb{R}^{d_x}$. After that, the agent recommends a slate $\boldsymbol{s} \in \mathcal{S}$, and then receives a binary reward $R \in \{0, 1\}$ and a list of $K$ binary ranks $[r_1, \ldots, r_K] \in \{0, 1\}^K$ that depend on both the context $\boldsymbol{x}$ and the slate $\boldsymbol{s}$. The reward $R$ indicates whether the user interacted with the slate $\boldsymbol{s}$ and for any $\ell \in [K]$ the rank $r_\ell$ indicates whether the user interacted with the $\ell$-th item in the slate, $s_\ell$. The user can interact with *at most* one item in the slate, and thus $R = \sum_{\ell \in [K]} r_\ell$. We let $\bar{R} = 1 - R$ so that $\bar{R} + \sum_{\ell \in [K]} r_\ell = 1$. Then the vector $(\bar{R}, r_1, \ldots, r_K) \in \mathbb{R}^{K+1}$ has one non-zero entry which is equal to one.

We assume that the context $\boldsymbol{x}$ decomposes into two vectors as $\boldsymbol{x} = (\boldsymbol{y}, \boldsymbol{z})$ where $\boldsymbol{y} \in \mathbb{R}^{d'}$ and $\boldsymbol{z} \in \mathbb{R}^{d_z}$. Here $\boldsymbol{y}$ denotes the engagement features that are useful for predicting the reward of a slate, independently of its items and the user interests. On the other hand, $\boldsymbol{z} \in \mathbb{R}^{d_z}$ denotes the remaining features in the context $\boldsymbol{x}$, which summarize the user interests. The dimensions of $\boldsymbol{z}$ and $\boldsymbol{x}$ are varying as they can contain the list of previously viewed items whose length may differ from one user to another. For this reason, these dimensions are subscripted by $\boldsymbol{z}$ and $\boldsymbol{x}$, respectively. In contrast, to simplify the notation, the dimension of $\boldsymbol{y}$, $d'$, is fixed (although it can also be varying). We give a summary of our notation in Table 2 in Appendix A.

## 3.2 Modeling Rank and Reward

As we highlighted before, engagement features can be strong predictors of the reward of a slate independently of the quality of its items. Thus a model using these features while discarding the user interests might enjoy a high likelihood. But such a model is useless for personalized recommendation as it does not learn the user interests. This observation is often used to justify abandoning likelihood-based approaches in favor of ranking. Instead, `PRR` solves this issue by carefully incorporating both the engagement features $\boldsymbol{y}$, the user interests features $\boldsymbol{z}$ and the whole slate $\boldsymbol{s}$ to predict interactions accurately. The vector $(\bar{R}, r_1, \ldots, r_K) \in \mathbb{R}^{K+1}$ has exactly one non-zero entry which is equal to one (Section 3.1). Thus we model it using a categorical distribution. Precisely, the `PRR` model has the following form

$$\bar{R}, r_1, \ldots, r_K | \boldsymbol{x}, \boldsymbol{s} \sim \text{cat}\left(\frac{\theta_0}{Z}, \frac{\theta_1}{Z}, \ldots, \frac{\theta_K}{Z}\right), \quad Z = \sum_{k=0}^{K} \theta_\ell, \tag{1}$$

where $\bar{R}, r_1, \ldots, r_K$ and $\boldsymbol{x} = (\boldsymbol{y}, \boldsymbol{z})$ are defined in Section 3.1, cat() is the categorical distribution, $\theta_0$ is the score of no interaction with the slate and $\theta_\ell$, for $\ell \in [K]$, is the score of interaction with the $\ell$-th item in the slate, $s_\ell$. The engagement features $\boldsymbol{y}$ are used to produce the positive score $\theta_0$ which is high if the chance of no interaction with the slate is high, independently of its items. It is defined as

$$\theta_0 = \exp(\boldsymbol{y}^\top \boldsymbol{\phi}), \tag{2}$$

where $\boldsymbol{\phi}$ is a vector of learnable parameters of dimension $d' > 0$. Similarly, let $\ell \in [K]$, the positive score $\theta_\ell$ is associated with the item in position $\ell$ in the slate, $s_\ell$, and is calculated in a way that captures user interests, position biases, and interactions that occur by *accident*. Precisely, given a slate $\boldsymbol{s} = (s_\ell)_{\ell \in [K]} = (s_1, \ldots, s_K)$ and user interests features $\boldsymbol{z}$, the score $\theta_\ell$ has the following form

$$\theta_\ell = \exp\{g_{\boldsymbol{\Gamma}}(\boldsymbol{z})^\top \boldsymbol{\Psi}_{s_\ell}\} \exp(\gamma_\ell) + \exp(\alpha_\ell). \tag{3}$$

Again this formulation is motivated by practitioners experience. The quantity $\exp(\alpha_\ell)$ denotes the additive bias for position $\ell \in [K]$ in the slate. It is high if there is a high chance of interaction with the $\ell$-th item in the slate irrespective of how appealing it is to the user. This quantity also explains interactions that are not associated at all with the recommendation (e.g., clicks by accident). The quantity $\exp(\gamma_\ell)$ is the multiplicative bias for position $\ell \in [K]$. It is high if a recommendation is *boosted* by being in position $\ell \in [K]$. To see this, consider the example of ad placement and assume that we recommend a large slate of the form (*phone,…, microphone*). Here *phone* is placed in the first position while *microphone* is placed in the last one. Now suppose that the user clicked on *phone*. Then from a ranking perspective, we would assume that the user prefers the *phone* over the *microphone*. However, the user might have clicked on the *phone* only because it was placed in the top position. `PRR` captures this through the multiplicative terms $\exp(\gamma_\ell)$.

Finally, the main quantity of interest is the recommendation score $g_{\mathbf{\Gamma}}(\boldsymbol{z})^{\top}\boldsymbol{\Psi}_a$ for $a \in [P]$, which can be understood as follows. The vector $\boldsymbol{z} \in \mathbb{R}^{d_z}$ represents the user interests and the parameter vector $\boldsymbol{\Psi}_{s_\ell} \in \mathbb{R}^d$ represents the embedding of the $\ell$-th item in the slate, $s_\ell$. The vector $\boldsymbol{z}$ is first mapped into a fixed size $d$-dimensional embedding space using $g_{\mathbf{\Gamma}}(\cdot)$. The resulting inner product $g_{\mathbf{\Gamma}}(\boldsymbol{z})^{\top}\boldsymbol{\Psi}_{s_\ell}$ produces a positive or negative score that quantifies how good $s_\ell$ is to the user with interests $\boldsymbol{z}$. In practice, $\boldsymbol{z}$ can be the sequence of previously viewed items, in which case $g_{\mathbf{\Gamma}}$ is a sequence model (Hidasi et al., 2015).

### 3.3 Learning

`PRR` has multiple parameters $\boldsymbol{\phi}, \mathbf{\Gamma}, \boldsymbol{\Psi}, \gamma_\ell$, and $\alpha_\ell$ for $\ell \in [K]$, which are learned using the maximum likelihood principle. Precisely, we assume access to logged data $\mathcal{D}_n$ of the form $\mathcal{D}_n = \{\boldsymbol{x}_i, \boldsymbol{s}_i, \bar{R}_i, r_{i,1}, \ldots, r_{i,K} ; i \in [n]\}$, such that $\boldsymbol{x}_i = (\boldsymbol{y}_i, \boldsymbol{z}_i)$ for any $i \in [n]$. Let $Z_i = \exp(\boldsymbol{y}_i^{\top}\boldsymbol{\phi}) + \sum_{\ell \in [K]} \exp\{g_{\mathbf{\Gamma}}(\boldsymbol{z}_i)^{\top}\boldsymbol{\Psi}_{s_{i,\ell}}\} \exp(\gamma_\ell) + \exp(\alpha_\ell)$ be the normalizing constant for the $i$-th data-point in $\mathcal{D}_n$, then log-likelihood reads

$$\mathcal{L}(\mathcal{D}_n; \boldsymbol{\phi}, \mathbf{\Gamma}, \boldsymbol{\Psi}, \boldsymbol{\gamma}, \boldsymbol{\alpha}) = \sum_{i \in [n]} \log P(\bar{R}_i, r_{i,1}, \ldots, r_{i,K} \mid \boldsymbol{x}_i, \boldsymbol{s}_i, \boldsymbol{\phi}, \mathbf{\Gamma}, \boldsymbol{\Psi}, \boldsymbol{\gamma}, \boldsymbol{\alpha}), \tag{4}$$

$$= \sum_{i \in [n]} (\boldsymbol{y}_i^{\top}\boldsymbol{\phi}) \, \mathbb{I}_{\{\bar{R}_i=1\}} + \sum_{\ell \in [K]} \left(\log \left(\exp\{g_{\mathbf{\Gamma}}(\boldsymbol{z}_i)^{\top}\boldsymbol{\Psi}_{s_{i,\ell}}\} \exp(\gamma_\ell) + \exp(\alpha_\ell)\right)\right) \mathbb{I}_{\{r_{i,\ell}=1\}} - \log(Z_i).$$

Finally, the parameters are estimated as $\hat{\boldsymbol{\phi}}_n, \hat{\mathbf{\Gamma}}_n, \hat{\boldsymbol{\Psi}}_n, \hat{\boldsymbol{\gamma}}_n, \hat{\boldsymbol{\alpha}}_n = \operatorname{argmax}_{\boldsymbol{\phi}, \mathbf{\Gamma}, \boldsymbol{\Psi}, \boldsymbol{\gamma}, \boldsymbol{\alpha}} \mathcal{L}(\mathcal{D}_n; \boldsymbol{\phi}, \mathbf{\Gamma}, \boldsymbol{\Psi}, \boldsymbol{\gamma}, \boldsymbol{\alpha})$. With a slight abuse of notation, we will refer to the learned parameters by $\boldsymbol{\phi}, \mathbf{\Gamma}, \boldsymbol{\Psi}, \boldsymbol{\gamma}, \boldsymbol{\alpha}$ in the sequel.

### 3.4 Decision Making

From Eq. (1), the probability of an interaction with the slate is

$$P(R = 1 \mid \boldsymbol{x}, \boldsymbol{s}) = 1 - P(\bar{R} = 1 \mid \boldsymbol{x}, \boldsymbol{s}) = 1 - \frac{\theta_0}{Z} = 1 - \frac{\theta_0}{\theta_0 + \sum_{\ell \in [K]} \theta_\ell}. \tag{5}$$

Then, from Eqs. (2) and (3), the decision rule follows as

$$\operatorname{argmax}_{\boldsymbol{s} \in \mathcal{S}} P(R = 1 \mid \boldsymbol{x}, \boldsymbol{s}) = \operatorname{argmin}_{\boldsymbol{s} \in \mathcal{S}} \frac{\theta_0}{\theta_0 + \sum_{\ell \in [K]} \theta_\ell} \overset{(i)}{=} \operatorname{argmax}_{\boldsymbol{s} \in \mathcal{S}} \sum_{\ell \in [K]} \theta_\ell,$$

$$= \operatorname{argmax}_{\boldsymbol{s} \in \mathcal{S}} \sum_{\ell \in [K]} \exp\{g_{\mathbf{\Gamma}}(\boldsymbol{z})^{\top}\boldsymbol{\Psi}_{s_\ell}\} \exp(\gamma_\ell) + \exp(\alpha_\ell)$$

$$\overset{(ii)}{=} \operatorname{argmax}_{\boldsymbol{s} \in \mathcal{S}} \sum_{\ell \in [K]} \exp\{g_{\mathbf{\Gamma}}(\boldsymbol{z})^{\top}\boldsymbol{\Psi}_{s_\ell}\} \exp(\gamma_\ell), \tag{6}$$

where $(i)$ and $(ii)$ follow from the fact that both $\theta_0$ and $\exp(\alpha_\ell)$ are additive and do not depend on $\boldsymbol{s}$. Our goal is to reduce decision-making to a MIPS task. Thus the parametric form $\boldsymbol{u}^{\top}\boldsymbol{\beta}$ must be satisfied, which means that the sum $\sum_{\ell \in [K]}$, the exponential in $\exp\{g_{\mathbf{\Gamma}}(\boldsymbol{z})^{\top}\boldsymbol{\Psi}_{s_\ell}\}$ and the term $\exp(\gamma_\ell)$ in Eq. (6) need to be removed. This is achieved by first sorting the position biases as

$$i_1, \ldots, i_K = \operatorname{argsort}(\gamma). \tag{7}$$

This is done once since Eq. (7) neither depend on the items nor the user. Then MIPS is performed as

$$s'_1, \ldots, s'_K = \operatorname{argsort}(g_{\mathbf{\Gamma}}(\boldsymbol{z})^{\top}\boldsymbol{\Psi})_{1:K}. \tag{8}$$

Finally, the recommended slate $\boldsymbol{s} = (s_1, s_2, \ldots, s_K)$ is obtained by rearranging the items $s'_1, \ldots, s'_K$ as

$$s_1, s_2, \ldots, s_K = s'_{i_1}, s'_{i_2}, \ldots, s'_{i_K}. \tag{9}$$

In other terms, we select the top-K items with the highest recommendation scores $g_{\mathbf{\Gamma}}(\boldsymbol{z})^{\top}\boldsymbol{\Psi}_a$ for $a \in [P]$. We then place the highest scoring item into the best position, that is the position $\ell \in [K]$ with the largest

value of $\gamma_\ell$. Then the second-highest scoring item is placed into the second-best position, and so on. The procedure in Eqs. (7) to (9) is equivalent to the decision rule in Eq. (6). It is also computationally efficient as Eq. (8) can be performed roughly in $\mathcal{O}(\log P)$ thanks to fast approximate MIPS algorithms (Malkov & Yashunin, 2018), while Eq. (6) requires roughly $\mathcal{O}(P^K)$. The time complexity is also improved compared to ranking approaches by $\mathcal{O}(P/\log P)$. This makes `PRR` scalable to huge action spaces.

Note that $\boldsymbol{\phi}, \boldsymbol{\alpha}$ are *nuisance* parameters as they are not needed for decision making; only the recommendation scores $g_{\boldsymbol{\Gamma}}(\boldsymbol{z})^\top \Psi_a$ and the multiplicative position biases $\exp(\gamma_\ell)$ are used in the procedure in Eqs. (7) to (9). While not used in decision-making, learning these parameters is necessary to accurately predict the recommendation scores. Also, including them in the model provides room for interpretability in some cases.

To summarize, `PRR` has the following properties. **1)** It models the joint distribution of the reward and ranks $(\bar{R}, r_1, \ldots, r_K)$ in the simple formulation given in Eq. (1). **2)** It makes use of engagement features $\boldsymbol{y}$ in order to help learn the recommendation signal more accurately. **3)** Its recommendation scores have a parametric form that is suitable for MIPS, which allows fast delivery of recommendations in $\mathcal{O}(\log P)$.

When compared to prior works, `PRR` enjoys the advantages of both worlds, reward and ranking approaches. Reward based approaches (Dudík et al., 2012) focus exclusively on the reward signal. This has a very profound advantage since what is optimized offline is aligned with the reward observed in A/B tests. However, the rank signal is ignored, and this loss of information makes learning difficult, especially in large-scale tasks. On the other hand, ranking approaches (Rendle et al., 2012) are driven by heuristics focused on proxy scores for individual items, which are not aligned with A/B test results (Garcin et al., 2014). `PRR` is similar to reward-based approaches as it directly optimizes the reward as measured in A/B tests. It is also similar to ranking as it makes use of the rank signal. However, `PRR` optimizes the reward for the whole slates and incorporates extra factors that may influence the reward independently of the quality of recommendations.

# 4 EXPERIMENTS

We evaluate `PRR` by creating synthetic and real-world problems that mimic the sequential interactions between users and recommender systems. The other evaluation alternatives consist in either using information-retrieval metrics or IPS. Unfortunately, the former is not aligned with online A/B test results, while the latter can suffer high bias and variance in large-scale settings (Aouali et al., 2022). Next we briefly present our experimental design while we defer further detail to Appendix B.

## 4.1 Baselines

`PRR` optimizes the reward offline using logged data and thus we only compare it to off-policy reward optimizing approaches. This does not include collaborative filtering, content-based (Su & Khoshgoftaar, 2009; Lops et al., 2011) or on-policy reinforcement learning methods (e.g., (Ie et al., 2019)). Thus we mainly compare `PRR` to the methods reviewed in Sections 2.1 and 2.2. This includes IPS and its variants (Section 2.1) and three DMs (Section 2.2) derived from `PRR` which are used to validate some of our modeling assumptions. We do not include other direct methods because the variants of `PRR` should be preferred as they are specifically designed for our problem. The others are either designed for single-item recommendation (slates of size 1) or for information-retrieval problems that are different from ours such as web search.

**Variants of PRR:** we consider three variants of `PRR`. First, `PRR-reward` uses only the reward and ignores the rank. `PRR-reward` is trained on both, successful and unsuccessful slates. Second, `PRR-rank` only uses the rank and is consequently trained on successful slates only. Finally, `PRR-bias` ignores the engagement features $\boldsymbol{y}$ and sets $\theta_0 = \exp(\phi)$ where $\phi$ is a scalar parameters ($\phi$ replaces $\boldsymbol{y}^\top \boldsymbol{\phi}$). Comparing `PRR` to `PRR-reward` and `PRR-rank` is to show the benefits of combining both signals, while comparing it to `PRR-bias` it to highlight

the effect and importance of the engagement features $\boldsymbol{y}$. The three models are summarized below.

$$\texttt{PRR-reward:} \qquad \bar{R}, r_1, \ldots, r_K \mid \boldsymbol{x}, \boldsymbol{s} \sim \operatorname{cat}\Big(\frac{\theta_0}{Z}, \frac{\sum_{\ell=1}^{K} \theta_\ell}{Z}\Big), \quad Z = \sum_{\ell=0}^{K} \theta_\ell \,,$$

$$\texttt{PRR-rank:} \qquad r_1, \ldots, r_K \mid \boldsymbol{x}, \boldsymbol{s} \sim \operatorname{cat}\Big(\frac{\theta_1}{Z}, \ldots, \frac{\theta_K}{Z}\Big), \quad Z = \sum_{\ell=1}^{K} \theta_\ell \,,$$

$$\texttt{PRR-bias:} \qquad \bar{R}, r_1, \ldots, r_K \mid \boldsymbol{x}, \boldsymbol{s} \sim \operatorname{cat}\Big(\frac{\phi}{Z}, \frac{\theta_1}{Z}, \ldots, \frac{\theta_K}{Z}\Big), \quad Z = \phi + \sum_{\ell=1}^{K} \theta_\ell.$$

where $\theta_0$ and $\theta_\ell$ for $\ell \in [K]$ are defined in Eqs. (2) and (3) while $\phi \in \mathbb{R}$ in $\texttt{PRR-bias}$ is a learnable parameter.

**Inverse propensity scoring:** We also consider IPS estimators of the expected reward of policies that are designed by removing the preference bias of the logging policy $\pi_0$ in data $\mathcal{D}_n$. This is achieved by re-weighting samples using the discrepancy between the learning policy $\pi$ and the logging policy $\pi_0$ such as

$$\hat{V}_n^{\mathrm{IPS}}(\pi) = \frac{1}{n} \sum_{i=1}^{n} R_i \frac{\pi(\boldsymbol{s}_i|\boldsymbol{z}_i)}{\pi_0(\boldsymbol{s}_i|\boldsymbol{z}_i)} \,. \tag{10}$$

This estimator is unbiased when $\pi$ and $\pi_0$ have common support. But it can be highly biased when this assumption is violated (Sachdeva et al., 2020), which is common in practice. It also suffers high variance. One way to mitigate this is to reduce the action space from slates to items (Li et al., 2018). This is achieved by assuming that the reward $R$ is the sum of rank $r_1, \ldots, r_K$, and that the $\ell$-th rank, $r_\ell$, only depends on the item $s_\ell$ and its position $\ell$. This allows estimating the expected reward of the learning policy $\pi$ as

$$\hat{V}^{\mathrm{IIPS}}(\pi) = \frac{1}{n} \sum_{i=1}^{n} \sum_{\ell=1}^{K} r_{i,\ell} \frac{\pi(s_{i,\ell}, \ell|\boldsymbol{z}_i)}{\pi_0(s_{i,\ell}, \ell|\boldsymbol{z}_i)} \,, \tag{11}$$

where $\pi(a, \ell|\boldsymbol{z})$ and $\pi_0(a, \ell|\boldsymbol{z})$ are the marginal probabilities that the learning policy $\pi$ and the logging policy $\pi_0$ place the item $a$ in position $\ell \in [K]$ given user interests $\boldsymbol{z}$, respectively. Note that in practice computing these marginals is often intractable for both $\pi$ and $\pi_0$; in which case approximation must be employed.

The next step is to optimize the estimator ($\hat{V}_n^{\mathrm{IPS}}(\pi)$ or $\hat{V}_n^{\mathrm{IIPS}}(\pi)$) to find the policy that will be used for decision-making (Swaminathan & Joachims, 2015). To achieve this, we need to parameterize the learning policy $\pi$. Here we assume that $\pi$ is parametrized as a factored softmax

$$\pi(\boldsymbol{s} \mid \boldsymbol{z}) = \pi_{\Xi, \beta, K}(\boldsymbol{s}|\boldsymbol{z}) = \prod_{\ell=1}^{K} p_{\Xi, \beta}(s_\ell|\boldsymbol{z}) \,, \qquad p_{\Xi, \beta}(a|\boldsymbol{z}) = \frac{\exp\{f_{\Xi}(\boldsymbol{z})^\top \beta_a\}}{\sum_{a' \in [P]} \exp\{f_{\Xi}(\boldsymbol{z})^\top \beta_{a'}\}} \,, \tag{12}$$

where $\beta_a \in \mathbb{R}^d$ is the embedding of item $a$ and $f_{\Xi}$ maps user interests $\boldsymbol{z} \in \mathbb{R}^{d_z}$ into a $d$-dimensional embedding. Finally, $\hat{V}_n^{\mathrm{IIPS}}(\pi)$ also requires the marginal probabilities $\pi(a, \ell|\boldsymbol{z})$ and $\pi_0(a, \ell|\boldsymbol{z})$. In our case, $\pi(a, \ell|\boldsymbol{z}) = p_{\Xi, \beta}(a|\boldsymbol{z})$ while we may need to approximate $\pi_0(a, \ell|\boldsymbol{z})$ depending on the logging policy. While convenient, factored policies have significant limitations. In particular, IIPS with factored policies might cause the learned policy to converge to selecting slates with repeated items, which is illegal. Thus to be fair to IIPS, we use sampling *without replacement* in decision making. Another alternative to mitigate this is the top-K heuristic (Chen et al., 2019) which causes the probability mass in $\pi_{\Xi, \beta, K}(\boldsymbol{s}|\boldsymbol{z})$ to be spread out over the top-K high scoring items rather than a single one. We denote the IIPS estimator combined with the top-K heuristic by $\texttt{top-K IIPS}$.

## 4.2 Synthetic Problems

We design a simulated A/B test protocol that takes different recommender systems as input and outputs their respective reward. We first define the problem instance consisting of the true parameters (oracle) and the logging policy as $\{\phi, \Psi, \gamma, \alpha, g_\Gamma(\cdot), P_{\boldsymbol{y}}(\cdot), P_{\boldsymbol{z}}(\cdot), P_K(\cdot)\}$ and $\pi_0$. Here $P_{\boldsymbol{y}}(\cdot), P_{\boldsymbol{z}}(\cdot)$, and $P_K(\cdot)$ are the

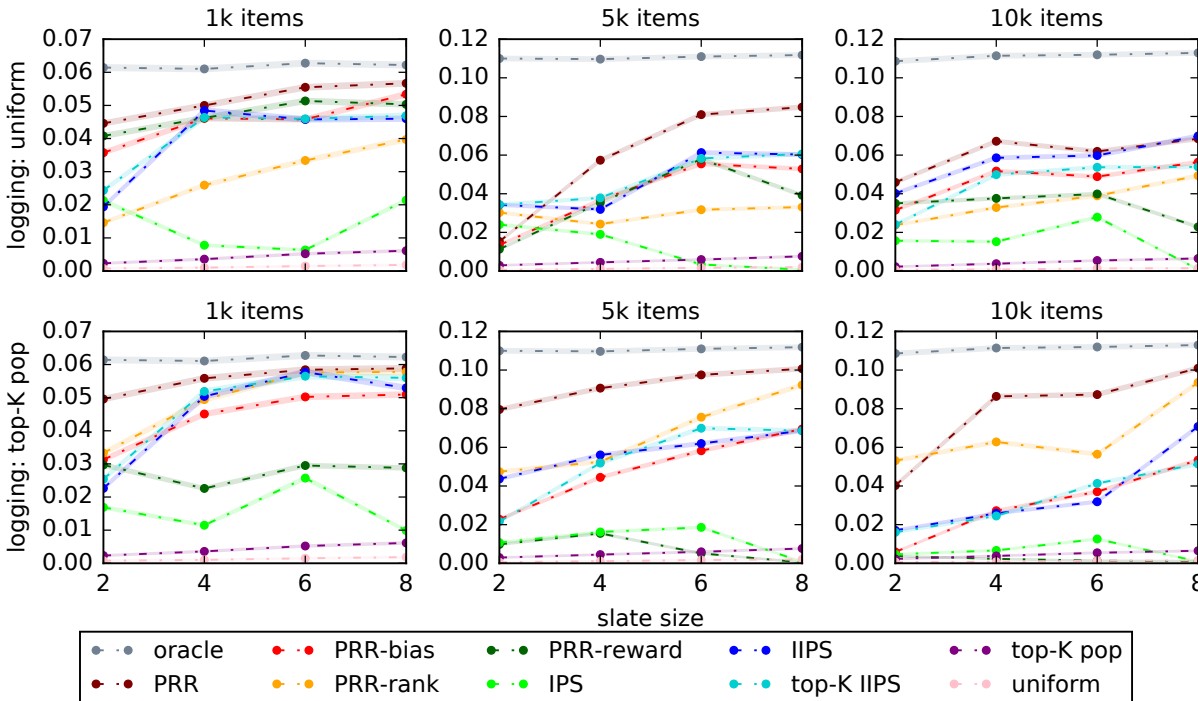

**Figure 2:** The reward (y-axis) of `PRR` and baselines in **synthetic problems** with varying slate sizes (x-axis), number of items (columns) and logging policies (rows). The shaded areas represent uncertainty and they are small since we run long A/B tests with $n_{\text{test}} = 100\text{k}$.

distributions of the engagement features, the user interests features and the slate size, respectively. Given the oracle, we produce offline training logs and propensity scores $\{\mathcal{D}, \mathcal{P}\}$ by running the logging policy $\pi_0$ in our simulated environment and observing its reward and rank. These logs are then used to train `PRR` and the baselines. After training, a simulated A/B test is used for testing. We defer a detailed description of our simulation environment for reproducibility to Appendix B.1. For instance, we summarize in Algorithm 1 the data generation process while we present in Algorithm 2 the simulated A/B test.

We consider two non-personalized logging policies. **(a) `uniform`:** this policy samples uniformly without replacement $K$ items from the catalog $[P]$. That is $\pi_0(\boldsymbol{s} \mid \boldsymbol{z}) = \frac{1}{P(P-1)\ldots(P-K+1)}$ for any slate $\boldsymbol{s} \in \mathcal{S}$ and any user interests $\boldsymbol{z}$. The marginal distribution can be computed in closed-form as $\pi_0(a, \ell|\boldsymbol{z}) = 1/P$. **(b) `top-K pop`:** this policy samples without replacement $K$ items where the probability of an item $a$ is proportional to the $L_2$ norm of its embedding, $\|\boldsymbol{\Psi}_a\|$. This is based on the intuition that a large value of $\|\boldsymbol{\Psi}_a\|$ means that item $a$ is recommended more often and thus it is more popular. We stress that this logging policy has access to the true embeddings $\boldsymbol{\Psi}$ of the simulated environment (Algorithm 1). Here the marginal distribution is intractable and we simply approximate is as $\pi_0(a, \ell|\boldsymbol{z}) \approx \|\Psi_a\|/\sum_{a'} \|\Psi_{a'}\|$ for IIPS.

In Fig. 2, we report the average A/B test reward of `PRR` with varying slate sizes, number of items and logging policies using 100k training samples. Overall, we observe that `PRR` outperforms the baselines across the different settings. Next we summarize the general trends of algorithms.

(a) **Varying logging policy:** models that use the reward only, `IPS` and `PRR-reward`, favor uniform logging policies while those that use only the rank, `IIPS` and `PRR-rank` perform better with the `top-K pop` logging policy. `PRR-bias` discards the slate-level features $\boldsymbol{y}$ and uses a single parameter $\phi$ for all slates. Thus `PRR-bias` benefits from uniform logging policies as they allow learning $\phi$ that works well across all slates. Indeed, in Fig. 2 the gap between `PRR` and `PRR-bias` shrinks for the uniform logging policy. Finally, the performance of `PRR` is relatively stable for both logging policies.

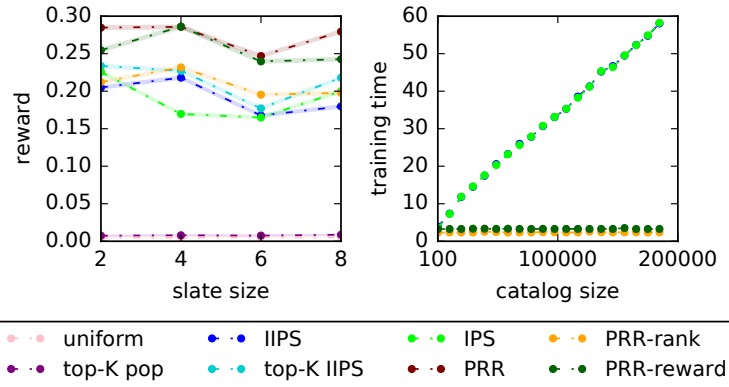

**Figure 3: On the left-hand side**, we report the reward (y-axis) of `PRR` and baselines in **session completion problems** with varying slate sizes (x-axis). **On the right-hand side**, we report the training time (y-axis) of `PRR` and baselines in **session completion problems** with varying catalog sizes (x-axis).

    **(b) Varying slate size:** the performance of models that use the reward only, `IPS` and `PRR-reward`, deteriorates when the maximum slate size increases. On the other hand, those that use only the rank, `IIPS` and `PRR-rank`, benefit from larger slates as this leads to displaying more item comparisons. The addition of the `top-K` heuristic improves the performance of `IIPS` in some cases by spreading the mass over different items, making it not only focus on retrieving one but several high scoring items. However, the increase in performance is not always guaranteed which might be due to our choice of hyperparameters or our approximation of the marginal distributions of policies. Finally, `PRR` performs well across all slate sizes as it uses both the reward and rank.

    **(c) Varying number of items:** the models that use the rank benefit from large slates. Here we observe that the increase in performance is more significant for large catalogs. In contrast, models that use only the reward suffer a drop in performance when the number of items increases.

## 4.3   Session Completion Problems

We use the Twitch dataset (Rappaz et al., 2021) to evaluate `PRR` on user session completion tasks. Roughly speaking, we process the dataset such that each user $u$ has a list $\mathcal{I}_u$ that contains the items that the user interacted with. We randomly split these user-item interactions $\mathcal{I}_u$ into two parts, $\mathcal{I}_u^{\text{VIEW}}$ and $\mathcal{I}_u^{\text{HIDE}}$. $\mathcal{I}_u^{\text{VIEW}}$ is observed while $\mathcal{I}_u^{\text{HIDE}}$ is hidden and should be predicted. That is, the baselines are evaluated based on their ability to predict the complete session of a user $\mathcal{I}_u$ by only observing a part of it, $\mathcal{I}_u^{\text{VIEW}}$.

Logged data $\mathcal{D}_n$ is collected using the `top-K pop` logging policy. In each iteration $i \in [n]$, we randomly sample a user $u_i$. Then, we recommend a slate $\boldsymbol{s}_i = (s_{i,\ell})_{\ell \in [K]}$ by sampling without replacement $K$ items with probabilities proportional to their popularity, i.e., their number of occurrences in the dataset. After recommending $\boldsymbol{s}_i = (s_{i,\ell})_{\ell \in [K]}$, we construct a binary vector $b_i = (\mathbb{I}_{\{s_{i,\ell} \in \mathcal{I}_u^{\text{HIDE}}\}})_{\ell \in [K]} = (\mathbb{I}_{\{s_{i,1} \in \mathcal{I}_u^{\text{HIDE}}\}}, \mathbb{I}_{\{s_{i,2} \in \mathcal{I}_u^{\text{HIDE}}\}}, \ldots, \mathbb{I}_{\{s_{i,K} \in \mathcal{I}_u^{\text{HIDE}}\}}) \in \mathbb{R}^K$. In other words, for any $\ell \in [K]$, $b_{i,\ell} = 1$ if the $\ell$-th item in the slate, $s_{i,\ell}$, is in the hidden user-item interaction $\mathcal{I}_u^{\text{HIDE}}$, and $b_{i,\ell} = 0$ otherwise. This binary vector $b_i$ is then used to generate the reward and rank signals $R_i, r_{i,1}, \ldots, r_{i,K}$ for user $u_i$ and slate $\boldsymbol{s}_i$. This allows constructing logged data $\mathcal{D}_n$. After training the baselines on $\mathcal{D}_n$, they are evaluated following the data collection process except that they make decisions instead of the logging policy $\pi_0$. More specific details about the data generation and testing processes are given in Appendix B.2.

In Fig. 3 (left-hand side), we report the results of `PRR` and the baselines on user session completion tasks. Note that there are no engagement features in this problem. Thus `PRR` is the same as `PRR-bias` and hence we only include `PRR` in Fig. 3. Overall, `PRR` outperforms the other baselines in the session completion problem. We also observe that `PRR-reward` has good performance in this scenario, while all the other methods have comparable performance which is lower than that of `PRR-reward`.

**Table 1:** Properties of `PRR` and the baselines.

| Method | Computational efficiency | Statistical efficiency |
|---|:---:|:---:|
| `PRR` | $\mathcal{O}(K)$ | High |
| `PRR-bias` | $\mathcal{O}(K)$ | Medium |
| `PRR-rank` | $\mathcal{O}(K)$ | Medium |
| `PRR-reward` | $\mathcal{O}(K)$ | Low |
| `IPS` | $\mathcal{O}(P)$ | Low |
| `IIPS` | $\mathcal{O}(P)$ | Medium |
| `Top-K IIPS` | $\mathcal{O}(P)$ | Medium |

### 4.4 Computational Efficiency

We assess the training speed of the algorithms with respect to the catalog and slate sizes $P$ and $K$. First, `PRR` and its variants compute $K + 1$ scores $\theta_0, \ldots, \theta_K$ and normalize them using $Z = \sum_{\ell=0}^{K} \theta_\ell$. Therefore, evaluating `PRR` and its variants in one data-point costs roughly $\mathcal{O}(K)$, where we omit the cost of computing the scores since it is the same for all algorithms. In contrast, `IPS` and its variants compute a softmax over the catalog. This requires computing the normalization constant $\sum_{a' \in [P]} \exp\{f_{\Xi}(z)^\top \beta_{a'}\}$ in (12). Thus the evaluation cost of `IPS` and its variants is roughly $\mathcal{O}(P)$. This is very costly compared to $\mathcal{O}(K)$ in realistic settings where $P \gg K$. An additional consideration to compare the training speed of algorithms is whether they use successful slates only, which significantly reduces the size of training data. Taking this into account, the fastest of all algorithms is the `PRR-rank` since its evaluation speed is $\mathcal{O}(K)$ and it is trained on successful slates only. In our experiments, `PRR-rank` is $\approx \mathbf{20\times}$ faster to train than `IPS`. These computational dependencies on $K$ and $P$ are also highlighted in Fig. 3 (right-hand side). In particular, `IPS` training time scales linearly in $P$ while `PRR` do not have such a dependency on $P$.

### 4.5 Limitations of `PRR`

After presenting and evaluating our algorithm, we are in a position to discuss its limitations. **(a) Modeling capacity:** `PRR` is trained to predict the reward of any slate which is a complex task. Thus high-dimensional embeddings might be needed to make accurate and calibrated predictions. As a result, the MIPS task produced by `PRR` might require high dimensional embeddings that do not conform to engineering constraints. A possible path to mitigate this is to observe that accurately predicting the reward is sufficient but not necessary for the recommendation task. When recommending, we only need to find the best slate. This is a simpler task and might be achieved with much smaller embeddings than those required by `PRR`. Therefore, one can train `PRR` with high-dimensional embeddings. Then optimize policies parametrized with low-dimensional embeddings using the learned reward estimates of `PRR` instead of IPS. The learned policy that fitst the engineering constraints will be then used in decision making. **(b) Theoretical analysis:** one of the main appeals of IPS is that it can be analyzed theoretically. This is difficult for `PRR` although a Bayesian analysis where we assume that the model of the environment is the same as that of `PRR` might be possible.

## 5 CONCLUSION

We present `PRR`, a scalable probabilistic model for personalized slate recommendation. `PRR` efficiently estimates the probability of a slate being successful by combining the reward and rank signals. It also optimizes the reward of the whole slate by distinguishing between slate-level and item-level features. Experiments attest that `PRR` outperforms the baselines and it is far more scalable, in both training and decision making. The shortcomings of our approach are discussed in Section 4.5 and we leave addressing them to future works.

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

## A  Summary of Notation

We provide a summary of our notation in Table 2.

**Table 2:** Notation.

| Notation | Definition |
|---|---|
| $\boldsymbol{x} = (\boldsymbol{y}, \boldsymbol{z}) \in \mathbb{R}^{d_x}$ | context. |
| $\boldsymbol{y} \in \mathbb{R}^{d'}$ | engagement features. |
| $\boldsymbol{z} \in \mathbb{R}^{d_z}$ | user interests features. |
| $R \in \{0, 1\}$ | reward indicator. |
| $\bar{R} \in \{0, 1\}$ | regret indicator. |
| $r_\ell \in \{0, 1\}$ | rank indicator of the item in position $\ell \in [K]$. |
| $\boldsymbol{\phi} \in \mathbb{R}^{d'}$ | engagement parameters. |
| $\gamma_\ell \in \mathbb{R}$ | multiplicative position bias in position $\ell \in [K]$. |
| $\alpha_\ell \in \mathbb{R}$ | additive position bias in position $\ell \in [K]$. |
| $g_{\boldsymbol{\Gamma}}(\boldsymbol{z}) \in \mathbb{R}^d$ | user embedding. |
| $\boldsymbol{\Psi} \in \mathbb{R}^{P \times d}$ | items embeddings. |
| $\theta_0 \in \mathbb{R}$ | score for no-interaction with the slate. |
| $\theta_\ell \in \mathbb{R}$ | score for an interaction with the item in position $\ell \in [K]$. |
| $\boldsymbol{s} = (s_1, ..., s_K)$ | slate of $K$ recommendations $s_\ell \in [P]$ for any $\ell \in [K]$. |

## B  Experimental Design

Here we give more information about our experiments. We start with the synthetic problems in Appendix B.1 and then present the session completion problems in Appendix B.2.

### B.1  Synthetic Problems

We design a simulated A/B test protocol that takes different recommender systems as input and outputs their respective reward. We first define the problem instance consisting of the true parameters (oracle) and the logging policy as $\{\boldsymbol{\phi}, \boldsymbol{\Psi}, \boldsymbol{\gamma}, \boldsymbol{\alpha}, g_{\boldsymbol{\Gamma}}(\cdot), P_{\boldsymbol{y}}(\cdot), P_{\boldsymbol{z}}(\cdot), P_K(\cdot)\}$ and $\pi_0$. Here $P_{\boldsymbol{y}}(\cdot), P_{\boldsymbol{z}}(\cdot)$ and $P_K(\cdot)$ are the distributions of the engagement features, the user interests features and the slate size, respectively. Given the oracle, we produce offline training logs and propensity scores $\{\mathcal{D}, \mathcal{P}\}$ by running the logging policy $\pi_0$ as described in Algorithm 1. These logs are then used to train PRR and competing baselines. After training, the simulated A/B test in Algorithm 2 is used for testing.

In all our experiments, the true parameters are sampled randomly as

$$\boldsymbol{\phi} \sim \mathcal{N}(\mu_\phi, \Sigma_\phi), \qquad \boldsymbol{\Psi} \sim \mathcal{N}(\mu_\psi, \Sigma_\psi), \qquad \boldsymbol{\Gamma} \sim \mathcal{N}(\mu_\Gamma, \Sigma_\Gamma), \qquad \boldsymbol{\gamma} \sim \mathcal{N}(\mu_\gamma, \Sigma_\gamma), \qquad \boldsymbol{\alpha} \sim \mathcal{N}(\mu_\alpha, \Sigma_\alpha).$$

For each user, the engagement features $\boldsymbol{y}$ are sampled randomly following the distribution $P_{\boldsymbol{y}} = \mathcal{N}(\mu_y, \Sigma_y)$. For the interest features $\boldsymbol{z}$, we assume that there are $L$ topics and $\boldsymbol{z}$ is consequently generated as follows. For each user $u$, we randomly sample the number of topics that interest user $u$ as $L_u \sim 1 + \mathcal{P}oison(3)$. After that, we uniformly sample $L_u$ topics that interest the user from $[L]$. It follows that $\boldsymbol{z} \in \mathbb{R}^L$ ($d_z = L$) is represented as a binary vector such as $z_\ell = 1$ if the user is interested in topic $\ell$, and $z_\ell = 0$ otherwise. For simplicity, the mapping $g_{\boldsymbol{\Gamma}}$ is linear and defined as $g_{\boldsymbol{\Gamma}}(\boldsymbol{z}) = \boldsymbol{\Gamma}\boldsymbol{z}$. Finally, for each user, the slate size is sampled uniformly from $[K]$ where $K$ is the maximum slate size. For reproducibility, the SEED is fixed at 42 which was chosen randomly. We considered realistic settings with relatively large catalogs and slates. The catalog size varies between 1000 and 10000 and the slate size varies between 2 and 8. PRR is suitable for larger catalogs but the baselines become very slow to train. This explains our choice of 10000 as a maximum catalog size. In optimization, we use Adam (Kingma & Ba, 2014) with a learning rate of 0.005 for 100 epochs using mini-batches of size 516.

---

**Algorithm 1:** Synthetic training logs

---

**Input:** oracle parameters $\{\boldsymbol{\phi}, \boldsymbol{\Psi}, \boldsymbol{\gamma}, \boldsymbol{\alpha}, g_{\boldsymbol{\Gamma}}(\cdot), P_{\boldsymbol{y}}(\cdot), P_{\boldsymbol{z}}(\cdot), P_K(\cdot)\}$, logging policy $\pi_0(\boldsymbol{s} \mid \boldsymbol{x})$, marginal
  logging policies $\pi_0(s_1|\boldsymbol{x}), \ldots, \pi_0(s_K|\boldsymbol{x})$, number of training samples $n_{\text{train}}$.
**Output:** logs $\mathcal{D}$, propensity scores $\mathcal{P}$.
$\mathcal{D} \leftarrow \{\,\}, \quad \mathcal{P} \leftarrow \{\,\}$
**for** $i = 1, \ldots, n_{\text{train}}$ **do**
  $\quad \boldsymbol{y}_i \sim P_{\boldsymbol{y}}(\cdot), \quad \boldsymbol{z}_i \sim P_{\boldsymbol{z}}(\cdot), \quad K_i \sim P_K(\cdot)$
  $\quad \boldsymbol{s}_i = (s_{i,1}, \ldots, s_{i,K_i}) \sim \pi_0(\cdot | \boldsymbol{z}_i)$
  $\quad \theta_0 \leftarrow \exp(\boldsymbol{y}_i^\top \boldsymbol{\phi})$
  $\quad$ **for** $\ell = 1, \ldots, K_i$ **do**
  $\quad\quad \theta_\ell \leftarrow \exp(g_{\boldsymbol{\Gamma}}(\boldsymbol{z}_i)^\top \boldsymbol{\Psi}_{s_{i,\ell}}) \exp(\gamma_\ell) + \exp(\alpha_\ell)$
  $\quad$ **end**
  $\quad \bar{R}_i, r_{i,1}, \ldots, r_{i,K} \sim \text{cat}\left(\frac{\theta_0}{Z}, \frac{\theta_1}{Z}, \ldots, \frac{\theta_K}{Z}\right), \quad Z = \sum_{\ell=0}^{K_i} \theta_\ell$
  $\quad \mathcal{D} \leftarrow \mathcal{D} \cup \{\boldsymbol{x}_i, \boldsymbol{s}_i, \bar{R}_i, r_{i,1}, \ldots, r_{i,K}\}$
  $\quad \mathcal{P} \leftarrow \mathcal{P} \cup \{\pi_0(\boldsymbol{s}_i | \boldsymbol{z}_i), \pi_0(s_{i,1}, 1 | \boldsymbol{z}_i), \ldots, \pi_0(s_{i,K}, K | \boldsymbol{z}_i)\}$
**end**

---

**Algorithm 2:** Synthetic A/B test

---

**Input:** oracle parameters $\{\boldsymbol{\phi}, \boldsymbol{\Psi}, \boldsymbol{\gamma}, \boldsymbol{\alpha}, g_{\boldsymbol{\Gamma}}(\cdot), P_{\boldsymbol{y}}(\cdot), P_{\boldsymbol{z}}(\cdot), P_K(\cdot)\}$, decision rules $d_{\text{A}}$ and $d_{\text{B}}$, number of
  testing samples $n_{\text{test}}$.
**Output:** lists of rewards $R_{\text{A}}$ and $R_{\text{B}}$.
$R_{\text{A}} \leftarrow \{\,\}, \quad R_{\text{B}} \leftarrow \{\,\}$
**for** $i = 1, \ldots, n_{\text{test}}$ **do**
  $\quad \boldsymbol{y}_i \sim P_{\boldsymbol{y}}(\cdot), \quad \boldsymbol{z}_i \sim P_{\boldsymbol{z}}(\cdot), \quad K_i \sim P_K(\cdot)$
  $\quad$ **for** $\text{M} \in \{A, B\}$ **do**
  $\quad\quad \boldsymbol{s}_i = (s_{i,1}, \ldots, s_{i,K_i}) \leftarrow d_{\text{M}}(\boldsymbol{y}_i, \boldsymbol{z}_i) \qquad$ (where $d_{\text{M}}$ is the decision rule of $\text{M} \in \{\text{A}, \text{B}\}$)
  $\quad\quad \theta_0 \leftarrow \exp(\boldsymbol{y}_i^\top \boldsymbol{\phi})$
  $\quad\quad$ **for** $\ell = 1, \ldots, K_i$ **do**
  $\quad\quad\quad \theta_\ell \leftarrow \exp(g_{\boldsymbol{\Gamma}}(\boldsymbol{z}_i)^\top \boldsymbol{\Psi}_{s_{i,\ell}}) \exp(\gamma_\ell) + \exp(\alpha_\ell)$
  $\quad\quad$ **end**
  $\quad\quad R_{\text{M}} \leftarrow R_{\text{M}} \cup \{1 - \frac{\theta_0}{Z}\}, \quad Z = \sum_{\ell=1}^{K_i} \theta_\ell$
  $\quad$ **end**
**end**

---

## B.2 Session Completion Problems

We use the Twitch dataset ([Rappaz et al., 2021](#)) to evaluate `PRR` on user session completion tasks. For each user, we randomly split the user-item interactions $\mathcal{I}_u$ into two parts, an observed part by the baselines $\mathcal{I}_u^{\text{VIEW}}$ and a hidden one $\mathcal{I}_u^{\text{HIDE}}$ that should be predicted. The task is to complete the observed user session $\mathcal{I}_u^{\text{VIEW}}$ to retrieve the whole session of a user $\mathcal{I}_u$. Logged data $\mathcal{D}_n$ is collected using the `top-K pop` logging policy as follows. In each iteration $i \in [n]$, we randomly sample a user $u_i$. Then, we recommend a slate $\boldsymbol{s}_i$ by sampling without replacement $K$ items with probabilities proportional to their popularity. Precisely, the probability of selecting an item $a$ is $c_a / \sum_{a'} c_{a'}$ where $c_a$ is the number of occurrences of item $a$ in the dataset. After that, we construct a binary vector $b_i = (\mathbb{I}_{\{s_{i,1} \in \mathcal{I}_{u_i}^{\text{HIDE}}\}}, \mathbb{I}_{\{s_{i,2} \in \mathcal{I}_{u_i}^{\text{HIDE}}\}}, \ldots, \mathbb{I}_{\{s_{i,K} \in \mathcal{I}_{u_i}^{\text{HIDE}}\}}) \in \mathbb{R}^K$. In other words, for any $\ell \in [K]$, $b_{i,\ell} = 1$ if the $\ell$-th item in the slate, $s_{i,\ell}$, is in the hidden user-item interaction $\mathcal{I}_u^{\text{HIDE}}$, and $b_{i,\ell} = 0$ otherwise. This binary vector $b_i$ is then used to generate the reward and rank signals as

$$\bar{R}_i, r_{i,1}, \ldots, r_{i,K} \sim \text{cat}\left(\frac{\beta_0 K}{\beta_0 K + \sum_{\ell \in [K]} \beta_\ell b_{i,\ell}}, \frac{\beta_1 b_{i,1}}{\beta_0 K + \sum_{\ell \in [K]} \beta_\ell b_{i,\ell}}, \ldots, \frac{\beta_K b_{i,K}}{\beta_0 K + \sum_{\ell \in [K]} \beta_\ell b_{i,\ell}}\right),$$

where $\beta_0$ and $\beta_\ell$ for $\ell \in [K]$ are sampled from $\mathcal{N}(3,9)$ and Uniform($[16]$), respectively. This allows us to generate a dataset in the form

$$\mathcal{D}_n = \left\{ \mathcal{I}_{u_i}^{\text{VIEW}}, \boldsymbol{s}_i, \bar{R}_i, r_{i,1}, \ldots, r_{i,K} \, ; i \in [n] \right\}.$$

Here $\mathcal{I}_{u_i}^{\text{VIEW}}$ can be seen as the user interest features. This data generation process is summarized in Algorithm 3.

---

**Algorithm 3:** Session completion training logs

---

**Input:** set of $U$ users, set of $P$ items, user-item interactions $\mathcal{I}_u^{\text{VIEW}}$ and $\mathcal{I}_u^{\text{HIDE}}$ for any user $u \in [U]$, number of occurrences $c_a$ for any item $a \in [P]$, maximum slate size $K$, position biases $\beta_\ell$ for $\ell \in [K]$, logging policy $\pi_0$, number of training samples $n_{\text{train}}$.

**Output:** logs $\mathcal{D}$, propensity scores $\mathcal{P}$.

$\mathcal{D} \leftarrow \{\,\}, \quad \mathcal{P} \leftarrow \{\,\}$

**for** $i = 1, \ldots, n_{\text{train}}$ **do**

    $u_i \sim \text{Uniform}([U])$

    $K_i \sim \text{Uniform}([K])$

    $\boldsymbol{s}_i = (s_{i,1}, \ldots, s_{i,K_i}) \sim \pi_0(\cdot \mid \mathcal{I}_{u_i}^{\text{VIEW}})$

    $b_i = (\mathbb{I}_{\{s_{i,1} \in \mathcal{I}_{u_i}^{\text{HIDE}}\}}, \mathbb{I}_{\{s_{i,2} \in \mathcal{I}_{u_i}^{\text{HIDE}}\}}, \ldots, \mathbb{I}_{\{s_{i,K} \in \mathcal{I}_{u_i}^{\text{HIDE}}\}})$

    $\bar{R}_i, r_{i,1}, \ldots, r_{i,K} \sim \text{cat}\left( \frac{\beta_0 K}{\beta_0 K + \sum_{\ell \in [K]} \beta_\ell b_{i,\ell}}, \frac{\beta_1 b_{i,1}}{\beta_0 K + \sum_{\ell \in [K]} \beta_\ell b_{i,\ell}}, \ldots, \frac{\beta_K b_{i,K}}{\beta_0 K + \sum_{\ell \in [K]} \beta_\ell b_{i,\ell}} \right)$

    $\mathcal{D} \leftarrow \mathcal{D} \cup \{\mathcal{I}_{u_i}^{\text{VIEW}}, \boldsymbol{s}_i, \bar{R}_i, r_{i,1}, \ldots, r_{i,K}\}$

    $\mathcal{P} \leftarrow \mathcal{P} \cup \{\pi_0(\boldsymbol{s}_i | \mathcal{I}_{u_i}^{\text{VIEW}}), \pi_0(s_{i,1}, 1 | \mathcal{I}_{u_i}^{\text{VIEW}}), \ldots, \pi_0(s_{i,K}, K | \mathcal{I}_{u_i}^{\text{VIEW}})\}$

**end**

---

After training the baselines on $\mathcal{D}_n$, they are evaluated following the data collection process except they are run instead of the logging policy $\pi_0$. More specific details about the data generation and testing processes are given in Appendix B.2.

---

**Algorithm 4:** Session completion A/B test

---

**Input:** set of $U$ users, set of $P$ items, user-item interactions $\mathcal{I}_u^{\text{VIEW}}$ and $\mathcal{I}_u^{\text{HIDE}}$ for any user $u \in [U]$, number of occurrences $c_a$ for any item $a \in [P]$, maximum slate size $K$, position biases $\beta_\ell$ for $\ell \in [K]$, logging policy $\pi_0$, number of testing samples $n_{\text{test}}$.

**Output:** lists of rewards $R_{\text{A}}$ and $R_{\text{B}}$.

$R_{\text{A}} \leftarrow \{\,\}, \quad R_{\text{B}} \leftarrow \{\,\}$

**for** $i = 1, \ldots, n_{\text{test}}$ **do**

    $u_i \sim \text{Uniform}([U])$

    $K_i \sim \text{Uniform}([K])$

    **for** $\text{M} \in \{A, B\}$ **do**

        $\boldsymbol{s}_i = (s_{i,1}, \ldots, s_{i,K_i}) \leftarrow d_{\text{M}}(\mathcal{I}_{u_i}^{\text{VIEW}}) \qquad (d_{\text{M}} \text{ is the decision rule of } \text{M} \in \{\text{A}, \text{B}\})$

        $b_i = (\mathbb{I}_{\{s_{i,1} \in \mathcal{I}_{u_i}^{\text{HIDE}}\}}, \mathbb{I}_{\{s_{i,2} \in \mathcal{I}_{u_i}^{\text{HIDE}}\}}, \ldots, \mathbb{I}_{\{s_{i,K} \in \mathcal{I}_{u_i}^{\text{HIDE}}\}})$

        $\bar{R}_i, r_{i,1}, \ldots, r_{i,K} \sim \text{cat}\left( \frac{\beta_0 K}{\beta_0 K + \sum_{\ell \in [K]} \beta_\ell b_{i,\ell}}, \frac{\beta_1 b_{i,1}}{\beta_0 K + \sum_{\ell \in [K]} \beta_\ell b_{i,\ell}}, \ldots, \frac{\beta_K b_{i,K}}{\beta_0 K + \sum_{\ell \in [K]} \beta_\ell b_{i,\ell}} \right)$

        $R_{\text{M}} \leftarrow R_{\text{M}} \cup \left\{ \frac{\sum_{\ell \in [K]} \beta_\ell b_{i,\ell}}{\beta_0 K + \sum_{\ell \in [K]} \beta_\ell b_{i,\ell}} \right\}$

    **end**

**end**

---

