# OpenReview forum: "Probabilistic Rank and Reward: A Scalable Model for Slate Recommendation"
_TMLR — Rejected by TMLR_

### Review · Reviewer_QqtH · 2023-06-03

**Summary Of Contributions:**

This paper presents a novel model, the Probabilistic Rank and Reward (PRR), for slate recommendations. Recognizing the increasing importance of recommender systems in various sectors, this work addresses the limitations of existing methods, such as the costly and time-consuming A/B testing, and the bias and variance problems in counterfactual estimators. The PRR model formalizes the slate recommendation scenario where a user is shown multiple items but can interact with only one. It utilizes two signals, reward and rank, that offer insight into user interests, and also incorporates slate and item-level features. Moreover, the model includes additional engagement features that enhance learning of user interests. The PRR's decision rule leverages maximum inner product search, facilitating fast decision-making for real-time systems. Synthetic and real-world experimental results demonstrate the proposed model's superior performance and scalability over a range of existing methods.


**Audience:**

Yes

**Broader Impact Concerns:**

I don’t have any broader impact concerns.

**Claims And Evidence:**

Yes

**Requested Changes:**

- I would suggest that the authors clarify the differences between the proposed method and those from [Ref A, B, C, D]. It would also be necessary to compare the proposed method with those from these related works in the experiments, or to elaborate on why it is not necessary.


- It would be beneficial to have experiments on a more practical setup such as with a richer backbone model like neural networks and doubly-robust estimation as a more powerful baseline. Applying weight clipping to the IPS variants to make these baselines more realistic would also be useful.


**Strengths And Weaknesses:**

**Strengths**
- The paper tackles practically important issues such as high variance and scalability in off-policy slate learning.


- The paper proposes a simple, yet effective solution (PRR) to the practical issues in off-policy slate learning.


- The paper is well-written, and the arguments are easy to follow.


- Experiments are done on both synthetic and real-world data, demonstrating the effectiveness and scalability of PRR against the baselines. In particular, the synthetic experiment design is comprehensive, providing insights as to when the proposed method is particularly superior.

**Weaknesses**
- Several most related papers are missing. In particular, [Ref A, B, C, D] seem very relevant and it should be necessary to distinguish the contributions from those in these papers. It would also be great to elaborate on why it is not necessary to compare PRR with the methods proposed in these papers.

  - [Ref A] Harrie Oosterhuis. Computationally Efficient Optimization of Plackett-Luce Ranking Models for Relevance and Fairness. SIGIR2021.

  - [Ref B] Harrie Oosterhuis. Learning-to-Rank at the Speed of Sampling: Plackett-Luce Gradient Estimation With Minimal Computational Complexity. SIGIR2022.
  - [Ref C] Otmane Sakhi, David Rohde, Alexandre Gilotte. Fast Offline Policy Optimization for Large Scale Recommendation. AAAI2023.


  - [Ref D] Matej Cief, Branislav Kveton, Michal Kompan. Pessimistic Off-Policy Optimization for Learning to Rank. arXiv2023. https://arxiv.org/abs/2206.02593



- There are no theoretical contributions in the paper, and the merits of the proposed methods are only verified in the experiments.


- The backbone model used in the experiment seems pretty simple, i.e., a linear model. How would the method work with more complex backbone models such as neural networks, which are more practical?


- The paper also compares the proposed method with some IPS variants, which are known to have very high variance in large action spaces. Therefore, weight clipping is often applied in practice to deal with this variance issue, but this seems to have been ignored in the experiments. It would also be interesting to compare the proposed method against doubly-robust methods, which are formulated in [Ref E] for example.

  - [Ref E] Haruka Kiyohara, Yuta Saito, Tatsuya Matsuhiro, Yusuke Narita, Nobuyuki Shimizu, Yasuo Yamamoto. Doubly Robust Off-Policy Evaluation for Ranking Policies under the Cascade Behavior Model. WSDM2022.

---

> ### Author Response · Authors · 2023-06-28
> **Response to Reviewer QqtH**
>
> We appreciate your feedback and review. We have responded to your comments individually. Please inform us if you have any additional questions or concerns.
>
> **Related work**
>
> **References [A, B, C]** introduce new _policy learning_ algorithms that assume the availability of a reward estimate $\hat{R}(x, s)$ for any context-slate pair $(x, s)$. These papers aim to find a policy $\pi$ that maximizes the expected reward. In other words, they seek to solve the optimization problem,
> $${\rm argmax}_\pi \ \frac{1}{n} \sum_i \sum_s \pi(s|x_i) \hat{R}(x_i, s)$$
> While a deterministic $\pi(\cdot|x)$ is theoretically optimal, in practice it is often approximated as a probability distribution to enable gradient-based optimization. Specifically, in the case of [A, B, C], $\pi(\cdot|x)$ is modeled using a Plackett-Luce distribution. The straightforward approach to solving this optimization problem involves using REINFORCE-style algorithms, which estimate gradients by sampling $s$ from $\pi(\cdot|x)$. However, naive REINFORCE can be computationally expensive due to the $\mathcal{O}(P)$ cost of sampling from $\pi(\cdot|x)$, where $P$ is the total number of items, and the high level of noise in the gradient estimates. Therefore, these papers propose methods to accelerate the optimization process.
>
> The contribution of [A, B] is the introduction of a lower variance gradient estimator, which assumes certain restrictive linear assumptions about the reward function. Although this estimator is less noisy, the iteration cost of the algorithm remains $\mathcal{O}(P)$. However, convergence with fewer iterations can be achieved in stochastic gradient descent (SGD) settings. On the other hand, [C] proposes an algorithm with an iteration cost of $\mathcal{O}(\log P)$ under the assumption that the item embeddings are pre-determined and fixed.
>
> It is important to note that all of these methods focus solely on optimizing the policy $\pi$ given an already available reward model $\hat{R},$ and their performance is contingent on the quality of this estimator. In contrast, PRR takes a different approach. It proposes a method for estimating $\hat{R}$ directly, bypassing the need for policy optimization, as demonstrated in Section 3.4.
>
> **References [D, E] and Clipped IPS** Roughly speaking, in comparison to our work, [D, E] address a distinct setting. Specifically, neither of these works models the scenario where the user is presented with the entire slate simultaneously and can interact with, at most, one item. This particular scenario is prevalent in domains such as online advertising, where users view a complete banner of products and, upon clicking on one item, the banner disappears, redirecting the user to a new page. Consequently, the models proposed in D and E do not incorporate this feature. To be precise, the dependent-click and position-based models described in Sections 5.2 and 5.3 of D allow multiple clicks on the slate, making them unsuitable for our setting. In contrast, the cascading models in D and E assume that the user interacts with items within a slate sequentially. That is, item $s_\ell$ at position $\ell \in [K]$ is examined only if item $s_{\ell-1}$ at the previous position is examined but not clicked. This differs from our setting, where all items are examined simultaneously (rather than sequentially) and the user can click on, at most, one of them.
>
> To strengthen our experiments, we have included click models such as cascading models (CM) and position-based models (PBM). The results can be found in this [anonymized link](https://github.com/anonymizeduser4020/tmlr_rebuttals/blob/main/rebuttals_baseline.pdf). While these click models exhibit good performance, PRR demonstrates even better performance. This is due to the fact that PRR is specifically designed for our setting, where the user views the entire slate and clicks on at most one item. Additionally, we also include improved IPS and IIPS with tuned clipping parameters. Despite the performance improvements observed with IPS and IIPS after clipping, PRR still outperforms them.
>
> **Backbone model** We acknowledge that the backbone models employed in our experiments are simple, and we recognize the significance of exploring more sophisticated architectures such as neural networks. However, the primary objective of our experiments was to validate the efficacy of precise reward modeling in accurately estimating the value of a slate for a given context when users interact with at most one item from a slate of $K$ items. Hence, our intention was to showcase this aspect independently from any numerical or optimization challenges. This rationale explains our choice of linear backbone models, which are commonly used in prior works on _reward-optimizing methods_. Adapting our method to complex neural networks like Transformers would necessitate additional considerations, which we believe exceed the scope of this study.

---

### Review · Reviewer_r1bK · 2023-06-15

**Summary Of Contributions:**

This paper proposes Probabilistic Rank and Reward (PRR), a scalable probabilistic model for personalized slate recommendation. The paper claims that both reward and ranking of items. Experiments on a synthetic dataset and a real dataset validates its efficiency.

**Audience:**

Yes

**Claims And Evidence:**

Yes

**Requested Changes:**

1. I suggest that the paper add some baselines, such as SlateQ and PRM.
2. Standard metrics are necessary.

**Strengths And Weaknesses:**

Strengths:
1.The formulation of slate reranking is clear.
2.Figure 1 is interesting.
3.The idea that combines the reward and the ranking is novel.

Weaknesses:
1. The paper considers a contextual bandit setting, which is limited. In practice, the behavior of the user is affected by the historical items interacted by the user.
2. The paper does not compare some start of art slate recommendation baselines, such as SlateQ, PRM.
3. The paper does not use normal metrics in ranking, such as NDCG and Hitrate.

---

> ### Author Response · Authors · 2023-06-29
> **Response to Reviewer r1bK**
>
> We sincerely appreciate your feedback and time. We have prepared point-by-point responses to your comments. If you have any additional questions or concerns, please let us know.
>
> **Baselines**
>
> PRR primarily focuses on optimizing the reward of slates offline. It is specifically designed for the common scenario where users examine the entire slate of recommendations at once and interact with, at most, a single item from that slate. Consequently, certain models were excluded from our experiments based on two main reasons.
>
> Firstly, some models do not optimize the reward offline but instead optimize alternative proxy metrics or learn the reward online (e.g., SlateQ). Previous studies have not directly compared methods that optimize the reward offline with those that learn it online or do not optimize it at all. Therefore, it is reasonable for us to concentrate on models that optimize the reward offline to enable a meaningful and relevant comparison.
>
> Secondly, the exclusion of other models from our experiments is due to their unsuitability for our specific setting, where users examine the entire slate and are limited to clicking on at most one item. These models allow users to click on multiple items within the slate, which does not align with our scenario. It is important to note that, for example, PRM considers a different problem; it relies on a base ranker and focuses on refining the item ordering within a slate, given by the base ranker. In contrast, PRR can be viewed as a potential base ranker for PRM when the user is presented with the entire slate and can only click on one item.
>
> To strengthen our experiments, we have included click models such as cascading models (CM) and position-based models (PBM) in our experiments. The results are found in this [anonymized link](https://github.com/anonymizeduser4020/tmlr_rebuttals/blob/main/rebuttals_baseline.pdf). While these click models exhibit good performance, PRR demonstrates even better performance. This is due to the fact that PRR is specifically designed for our setting, where the user views the entire slate and clicks on at most one item. Additionally, we also included in this [anonymized link](https://github.com/anonymizeduser4020/tmlr_rebuttals/blob/main/rebuttals_baseline.pdf) improved IPS and IIPS with tuned clipping parameters. Despite the performance improvements observed with IPS and IIPS after implementing clipping, PRR still outperforms them.
>
> In summary, our focus on optimizing the reward of slates offline and our specific design for the scenario where users examine the entire slate and interact with at most one item sets PRR apart from other models. We have also taken your suggestions into consideration and included additional experiments with click models, IPS, and IIPS with clipping, further confirming the appeal of PRR in our targeted setting.
>
> **Standard ranking metrics**
>
> Our experimental design allows us to directly measure the reward of PRR and other baselines on the test set. Thus the use of proxy ranking metrics is not relevant to our evaluation. This is because these metrics may not yield results that are sufficiently correlated with the actual reward, which we consider the golden standard for evaluation. Notably, [1] has demonstrated that using such metrics can lead to a significant disparity between offline evaluations and A/B test results. Finally, it is worth mentioning that previous studies on reward-optimizing algorithms for recommender systems (e.g., the contextual bandit literature) commonly use the test reward as a performance metric when available.
>
>
> [1] F. Garcin, B. Faltings, O. Donatsch, A. Alazzawi, C. Bruttin, and A. Huber. Offline and Online Evaluation of News Recommender Systems at Swissinfo.Ch. In Proc. of the 8th ACM Conference on Recommender Systems, RecSys ’14, pp. 169–176, 2014.

---

### Review · Reviewer_v7u1 · 2023-06-17

**Summary Of Contributions:**

The paper address the problem of slate recommendation by proposing a probabilistic rank and reward model (PRR) of user-item interaction data. The claimed contributions are as follows:

1. The problems of the gap between offline (observational) data and A/B test (interventional) data and high variance of the IPS family of estimators are addressed by proposing a joint probabilistic model of whether a user will engage with a slate of recommendations and if so, which item they are likely to pick.

2. Turning the rank and reward model into a slate of recommendations requires at least a linear search (or worse) through the item space if no further assumptions are made. Here, the authors take advantage of the linear structure in their proposed model to come up with an algorithm for slate recommendation based on maximum inner product search.

In doing so, they formulate the problem of slate recommendation as one modeling a user who picks only either zero or one items from any given slate, and that this can be modeled as a categorical distribution with an outside option (no-click) whose log probabilities are proportional to linear functions of user features and engagement. The engagement features allow the model to absorb possibly large variation in rewards for items and slate positions across sessions that are not directly due to the quality of recommendations. Experiments show the approach performs better and is more scalable than popularity and standard IPS approaches.

**Audience:**

Yes

**Claims And Evidence:**

No

**Requested Changes:**

1. Needs state-of-the-art baselines for comparison in the experiments, perhaps against methods already discussed in related work. This would give the required context for the results, both in terms of how PRR performs against other direct methods and what it gains in variance reduction by trading some bias.

2. Needs further discussion around whether PRR is a direct method, and further comparison to related direct methods, and more care taken with statistical biasedness and efficiency.

3. Suggesting to change the tone around fitting the nuisance variables in regression, as this is widely known and practiced.

**Strengths And Weaknesses:**

The paper provides a compelling solution to the real-world problem of slate recommendation in large action spaces. Its approach is well-formulated and motivated, offering helpful intuitions in the setup. The solution is straightforward, presenting simplicity in two dimensions. First, it focuses on instances where there is only one engagement with a slate, effectively reducing the problem space and accommodating many slate recommendation scenarios. Second, it uses a log-linear model with parameters that control user-item interactions and rewards based on position in the slate. The authors demonstrate that the recommendation setting, optimized for reward, can be formulated efficiently (log(n_items)) using a maximum inner product search.

However, the paper has several weaknesses. It lacks consideration of state-of-the-art baselines in the experimental section, focusing only on IPS and popularity baselines. It also fails to adequately position the proposed approach within the offline evaluation literature. There are unanswered questions concerning whether it's a direct method, the trade-offs of using PRR versus other models for evaluation, and how it performs experimentally against other direct method approaches. The paper does not address the known issue of bias in direct methods, which often occurs in recommendation datasets under model mismatch unless a doubly robust framework is employed. Furthermore, the authors claim that PRR is not just computationally efficient, but also statistically efficient. This assertion needs more rigorous substantiation. Lastly, the heavy emphasis on engagement covariates in the model seems overdone. It is standard practice to include nuisance covariates in linear models to absorb variation, so presenting it as a novel contribution in the context of recommendation appears to be misleading.

---

> ### Author Response · Authors · 2023-06-28
> **Response to Reviewer v7u1**
>
> Thank you for your feedback and time. We have addressed your comments point by point. Please let us know if you have any further questions or concerns.
>
> **Baselines**
>
> The main focus of PRR is to optimize the reward of slates offline. Also, PRR is specifically designed for the ubiquitous scenario where users examine the entire slate of recommendations and interact with, at most, a single item from the slate.
>
> Then, the exclusion of other models from our experiments is motivated by two reasons. Firstly, some models do not optimize the reward offline. Instead, they optimize alternative proxy metrics or learn the reward online. It is worth noting that previous studies have not directly compared methods that optimize the reward offline with those that learn it online or do not optimize it at all. Therefore, it is reasonable for us to focus on models that optimize the reward offline to provide a meaningful comparison. Secondly, other models that were not included in our experiments are not suitable for our specific setting, as they allow the user to click on multiple items within the slate. Our scenario revolves around the user examining the entire slate and clicking on at most one item.
>
> Finally, we have included click models such as cascading models (CM) and position-based models (PBM) in our experiments. The results are found in this [anonymized link](https://github.com/anonymizeduser4020/tmlr_rebuttals/blob/main/rebuttals_baseline.pdf). While these click models exhibit good performance, PRR demonstrates even better performance. This is due to the fact that PRR is specifically designed for our setting, where the user views the entire slate and clicks on at most one item. Additionally, we also included in this [anonymized link](https://github.com/anonymizeduser4020/tmlr_rebuttals/blob/main/rebuttals_baseline.pdf) improved IPS and IIPS with tuned clipping parameters. Despite the performance improvements observed with IPS and IIPS after implementing clipping, PRR still outperforms them.
>
> In summary, we focus on optimizing the reward of slates offline and PRR is specifically designed for the scenario where users examine the entire slate and interact with at most one item. We have also included additional experiments with click models, IPS, and IIPS with tuned clipping parameters, further confirming the appeal of PRR in our targeted setting.
>
> **PRR is a direct method**
>
> PRR is categorized as a direct method, as highlighted in Section 2.1, due to its model-based nature and the absence of the importance sampling technique employed in IPS. In PRR, we estimate the reward of a context-slate pair $(x, s)$ by training our reward model, as outlined in equation (1), using offline data.
>
> **Claims**
>
> We will provide the necessary clarifications regarding our contribution related to the use of nuisance parameters and differentiate between what is already known and what is novel. Indeed, the use of nuisance parameters is standard to some extent, but the separation between slate-level attributes and item-level ones is novel. In particular, we believe that our explicit utilization of engagement features to predict the score of no-interaction with the entire slate $(\theta_0)$ independently of the quality of its recommendations is a novel aspect. However, we would appreciate it if you could provide references to any related works that incorporate similar modeling assumptions in the context of recommender systems.
>
> Additionally, we acknowledge your concern regarding the term "statistical efficiency" potentially causing confusion. To ensure clarity, we will modify the terminology to "testing reward in our experiments". We used the term "statistical efficiency" to refer to the performance of algorithms in terms of testing reward in our experiments.

---

### Decision · Action_Editors · 2023-07-25

**Recommendation:** Reject

**Comment:**

The reviewers found the submission interesting, and also appreciated the efforts invested during the rebuttal period. In fact many of the reviewers' concerns were addressed during that period. Due to the remaning concerns (see below), the reviewers ultimately felt the paper falls just under the acceptance bar; specifically, 2 of the reviewers voted for weak reject, while 1 voted for weak accept.

The paper deinitely fits the TMLR scope, and given the reviewers' comments during the discussion period, I believe it can be ready for acceptance after incorporating the reviewers' suggestions (this was also mentioned by the reviewers). Unfortunately TMLR does not allow for a second round of reviews, therefore I **strongly encourage** the authors to resubmit to TMLR.

I am quoting below some of the points raised by the reviewers during the discussion period:
+ *"the experimental section is still lacking comparisons to state-of-the-art baselines. For a direct method approach, relying strongly on the modeling assumptions and hence low bias, there is more onus on experimental evaluation. Slate recommendation has been studied for several years at this point, so position-based engagement rates is still not strong enough a baseline to be convincing"* (Reviewer v7u1)
+ *"I still believe the claims of novelty around "engagement features to predict the score of no-interaction" should be attenuated. A prominent example of this being a widespread approach specifically in recommendation is in [Rendle, 2010]-- equation 1 specifies a first-order linear effect of covariates on engagement."* (Reviewer v7u1)
+ *"I would suggest adding the discussion regarding several relevant papers and also the clarification of the goal of the empirical study."* (Reviewer QqtH)


**Audience:**

Yes, it is definitely of interest for researchers working on the field.

**Claims And Evidence:**

Overall, the reviewers appreciated the rebuttal, which improved the paper and solved many of their concerns.

However, there are still a couple of outstanding issues. I am summarizing them here and providing more details further below:
+ The experimental section is still lacking comparisons to state-of-the-art baselines, which becomes more relevant for a direct method approach (Reviewers v7u1 and r1bK).
+ The claim of novelty around the "engagement features to predict the score of no-interaction" should be toned down, since it has appeared in the literature before (Reviewer v7u1).
+ The discussion of several relevant papers should be added, as well as clarification of the goal of the empirical study (Reviewer QqtH).


**Resubmission Of Major Revision:**

The authors may consider submitting a major revision at a later time.